# Expression and Activity of TRPA1 and TRPV1 in the Intervertebral Disc: Association with Inflammation and Matrix Remodeling

**DOI:** 10.3390/ijms20071767

**Published:** 2019-04-10

**Authors:** Takuya Kameda, Joel Zvick, Miriam Vuk, Aleksandra Sadowska, Wai Kit Tam, Victor Y. Leung, Kata Bölcskei, Zsuzsanna Helyes, Lee Ann Applegate, Oliver N. Hausmann, Juergen Klasen, Olga Krupkova, Karin Wuertz-Kozak

**Affiliations:** 1Institute for Biomechanics, ETH Zurich, Hoenggerbergring 64, 8093 Zurich, Switzerland; sa57384@cd6.so-net.ne.jp (T.K.); joel.zvick@hest.ethz.ch (J.Z.); vukm@student.ethz.ch (M.V.); aleksandra.sadowska@hest.ethz.ch (A.S.); kwuertz@ethz.ch (K.W.-K.); 2Department of Orthopaedic Surgery, Fukushima Medical University, 1 Hikarigaoka, Fukushima City, Fukushima 960-1295, Japan; 3Department of Orthopaedics and Traumatology, The University of Hong Kong, 21 Sassoon Road, Pokfulam, Hong Kong SAR, China; tamwk1@hku.hk (W.K.T.); vicleung@hku.hk (V.Y.L.); 4Department of Pharmacology and Pharmacotherapy, University of Pécs, H-7624 Pécs, Szigeti út 12., Hungary; kata.bolcskei@aok.pte.hu (K.B.); zsuzsanna.helyes@aok.pte.hu (Z.H.); 5János Szentágothai Research Centre, University of Pécs, H-7624 Pécs, Ifjúság útja 20., Hungary; 6Department of Musculoskeletal Medicine, Unit of Regenerative Therapy (UTR), University Hospital Lausanne, EPCR/02 Chemin des Croisettes 22, 1066 Epalinges, Switzerland; Lee.Laurent-Applegate@chuv.ch; 7Neuro- and Spine Center, St. Anna Hospital, Sankt-Anna-Strasse 32, 6006 Luzern, Switzerland; ohausmann@hin.ch; 8Clinic Prodorso, Walchestrasse 15, 8006 Zurich, Switzerland; info@prodorso.ch; 9Schön Clinic Munich Harlaching, Spine Center, Academic Teaching Hospital and Spine Research Institute of the Paracelsus Medical University Salzburg (AU), Harlachinger Str. 51, 81547 Munich, Germany; 10Department of Health Sciences, University of Potsdam, Am Neuen Palais 10, 14469 Potsdam, Germany

**Keywords:** low back pain, TRP channels, pro-inflammatory cytokines, aggrecanases, collagen, TRPA1, TRPV1, TRPV2, TRPV4, TRPC6

## Abstract

Transient receptor potential (TRP) channels have emerged as potential sensors and transducers of inflammatory pain. The aims of this study were to investigate (1) the expression of TRP channels in intervertebral disc (IVD) cells in normal and inflammatory conditions and (2) the function of Transient receptor potential ankyrin 1 (TRPA1) and Transient receptor potential vanilloid 1 (TRPV1) in IVD inflammation and matrix homeostasis. RT-qPCR was used to analyze human fetal, healthy, and degenerated IVD tissues for the gene expression of TRPA1 and TRPV1. The primary IVD cell cultures were stimulated with either interleukin-1 beta (IL-1β) or tumor necrosis factor alpha (TNF-α) alone or in combination with TRPA1/V1 agonist allyl isothiocyanate (AITC, 3 and 10 µM), followed by analysis of calcium flux and the expression of inflammation mediators (RT-qPCR/ELISA) and matrix constituents (RT-qPCR). The matrix structure and composition in caudal motion segments from TRPA1 and TRPV1 wild-type (WT) and knock-out (KO) mice was visualized by FAST staining. Gene expression of other TRP channels (A1, C1, C3, C6, V1, V2, V4, V6, M2, M7, M8) was also tested in cytokine-treated cells. TRPA1 was expressed in fetal IVD cells, 20% of degenerated IVDs, but not in healthy mature IVDs. TRPA1 expression was not detectable in untreated cells and it increased upon cytokine treatment, while TRPV1 was expressed and concomitantly reduced. In inflamed IVD cells, 10 µM AITC activated calcium flux, induced gene expression of IL-8, and reduced disintegrin and metalloproteinase with thrombospondin motifs 5 (ADAMTS5) and collagen 1A1, possibly via upregulated TRPA1. TRPA1 KO in mice was associated with signs of degeneration in the nucleus pulposus and the vertebral growth plate, whereas TRPV1 KO did not show profound changes. Cytokine treatment also affected the gene expression of TRPV2 (increase), TRPV4 (increase), and TRPC6 (decrease). TRPA1 might be expressed in developing IVD, downregulated during its maturation, and upregulated again in degenerative disc disease, participating in matrix homeostasis. However, follow-up studies with larger sample sizes are needed to fully elucidate the role of TRPA1 and other TRP channels in degenerative disc disease.

## 1. Introduction

Low back pain (LBP) is the leading cause of disability, activity limitation and lost productivity throughout the world today, with approximately 80% of all people suffering from back pain at least once in their life [1]. Degenerative disc disease (DDD), which is a progressive multifactorial disorder of the intervertebral disc (IVD), is an important factor that is involved in the development of LBP [1]. DDD is characterized by the release of inflammatory and catabolic mediators, including interleukin-1 beta (IL-1β), tumor necrosis factor alpha (TNF-α), prostaglandins, and proteases, which further promote the degradation of extracellular matrix (ECM) [2,3]. Pro-inflammatory cytokines IL-1β and TNF-α directly act as nociceptive triggers, but also activate the expression of other potentially nociceptive molecules including neuropeptides, interleukin-6 (IL-6), and interleukin-8 (IL-8) [4,5]. Molecular mechanisms involved in transduction and modulation of IL-1β and TNF-α signaling in DDD are not yet well-understood [6,7,8].

Signals that are provided by pro-inflammatory cytokines can be mediated via membrane channels [9,10]. Recently, transient receptor potential (TRP) channels have emerged as putative receptors for inflammation-associated molecules, positive/negative regulators of inflammation, and transducers of inflammatory pain [11,12,13]. TRP channels are cation selective transmembrane receptors with diverse physiological functions. Six families of mammalian TRP channels have been identified, classifying TRP channels according to their sequence homology and topological differences: TRPA (ankyrin), TRPC (canonical), TRPM (melastatin), TRPV (vanillin), TRPP (polycystin), and TRPML (mucolipin). Apart from TRPA, every subfamily has several members [14,15]. The dysregulation of TRP channels is implicated in many pathologies, including cardiovascular diseases, muscular dystrophies, and hyperalgesia [14,16]. Interestingly, the expression and activity of certain TRP channels is altered in painful joints and IVDs [11,12]. For example, the expression of TRPV4 in human IVDs was found to be elevated in regions of aggrecan depletion [17], while the gene expression of TRPC6 was associated with the severity of disc degeneration, increased expression of IL-6, and cell senescence [18,19]. TRPA1 and TRPV1 are involved in inflammation/nociception in sensory neurons and non-neuronal tissues [20,21,22]. TRPV1 is a non-selective calcium channel, the expression and activity of which increases after inflammatory stimulation in dorsal root ganglions (DRGs), possibly causing chronic hyperalgesia. TRPV1 also expressed in chondrocytes [23]. TRPA1 is a calcium permeable non-selective cation channel that is also widely expressed in sensory neurons and in non-neuronal cells, including chondrocytes [11]. TRPA1 is involved in various sensory and homeostatic functions, depending on the cell type [12,24]. TRPA1 and TRPV1 were shown to complement each other’s activities [21,25]. As numerous publications have linked TRPA1 and TRPV1 with inflammatory pain [21], therapeutic inhibition or the activation of TRPA1 and/or TRPV1 channels may be beneficial in the treatment of DDD. Therefore, the aims of this study were to investigate the (1) expression of TRP channels in IVD cells in normal and inflammatory conditions and (2) the function of TRPA1 and TRPV1 in disc inflammation and matrix homeostasis.

## 2. Results

### 2.1. Gene Expression of TRPA1 in Human IVD Tissue

Gene expression of TRPA1 was tested in human non-degenerated (*n* = 4) and degenerated (*n* = 20) IVD tissue in relation to disc region, disease type, pain score (for degenerated discs only), grade, and age. In the degenerated tissue, TRPA1 was found expressed in 20% of tested donors (four out of 20). Although only expressed in a subset of degenerated donors, TRPA1 was found in both annulus fibrosus (AF) and nucleus pulposus (NP), in an age range of 39–76 years, at pain scores 2 (= intense) and 3 (= disabling) and at Pfirrmann grades 2 and 3 (Table 1). TRPA1 was only expressed in one non-degenerated NP sample (in one out of four donors: male, 17 years old, grade 1, 2^−d*C*t^ = 4.57 × 10^−5^). Interestingly, TRPA1 was found expressed in cells isolated from human fetal IVD tissue (*n* = 4; 2^−d*C*t^ = 0.0463 ± 0.04742). It is known that TRPA1 can associate with TRPV1, thereby regulating its intrinsic properties independently of intracellular calcium. Table 1 summarizes the expression of TRPV1. A manuscript that provides details on the expression of TRPV1 in IVD tissue is currently in revision [26].

### 2.2. Gene Expression of TRPA1 and TRPV1 in Human IVD Cells Treated with Pro-Inflammatory Cytokines

TRPA1 and TRPV1 can be involved in modulating inflammation in both neuronal and non-neuronal cells [25]. Therefore, changes in the gene expression of TRPA1 and TRPV1 were tested in IVD cells stimulated with pro-inflammatory cytokines IL-1β and TNF-α (both 5 and 10 ng/mL) (*n* = 5) (Table 2). TRPA1 was under the detection limit in most untreated IVD cells, while its gene expression tended to increase with IL-1β treatment (*p* = 0.07 for IL-1 5 ng/mL) and it significantly increased with TNF-α (Figure 1A). TNF-α, but not IL-1β, significantly reduced gene expression of TRPV1 (Figure 1B). The induction of an inflammatory-catabolic shift upon cytokine treatment was demonstrated by an increase of IL-6, IL-8, cyclooxygenase-2 (COX-2), nerve growth factor (NGF), matrix metalloproteinase 1 (MMP1), matrix metalloproteinase 3 (MMP3), a disintegrin and metalloproteinase with thrombospondin motifs 4 (ADAMTS4), a disintegrin and metalloproteinase with thrombospondin motifs 5 (ADAMTS5), and a reduction in COL2A1. Tissue inhibitor of matrix metalloproteinase 1 (TIMP1), tissue inhibitor of matrix metalloproteinase 2 (TIMP2), Aggrecan and COL1A1 were unchanged (Appendix A). In IVD cells seeded in three-dimensional (3D) alginate beads and treated with IL-1β (5 ng/mL, 15 days, *n* = 4–10), gene expression of TRPA1 significantly increased on day 1, 8, and 15 (Figure 1C), while the gene expression of TRPV1 remained unchanged (Figure 1D). Immunostaining confirmed TRPA1 protein induction upon IL-1β treatment (5 ng/mL) (*n* = 3) (Figure 1E). Cell viability in alginate beads was monitored by Calcein/Ethidium homodimer staining and an average of 87% of living cells per treatment and time point was found (Appendix A).

### 2.3. Gene Expression of Other TRP Channels in Human IVD Cells Treated with Pro-Inflammatory Cytokines

Changes in the gene expression of other TRP channels, namely TRPC1, TRPC3, TRPC6, TRPV2, TRPV4, TRPV6, TRPM2, TRPM7, and TRPM8, were also tested in IVD cells upon cytokine stimulation (IL-1β and TNF-α, both 5 and 10 ng/mL) (*n* = 5). IL-1β significantly induced gene expression of TRPV4 and reduced TRPC6. TNF-α significantly activated the gene expression of TRPV2. The expression of other TRP channels showed no clear association with a pro-inflammatory treatment in the tested experimental settings (Figure 2A–G). Gene expression of TRPM2 and TRPM8 was under the detection limit in all of the treatment groups.

### 2.4. Functional Analysis of TRPA1 and TRPV1 in IVD Cells

Allyl isothiocyanate (AITC, 3 and 10 μM), which is an agonist for TRPA1 and TRPV1, was used to test the involvement of these channels in (1) inflammation responses and (2) matrix homeostasis within the IVD compartment. The activity of AITC was tested by Calcium flux assay (*n* = 3). AITC was applied in untreated IVD cells (expressing TRPV1) as well as in cells that were treated with pro-inflammatory cytokines (expressing TRPA1 and TRPV1). AITC did not induce calcium flux in the untreated IVD cells. AITC significantly induced calcium flux in both IL-1β and TNF-α-treated cells, suggesting the involvement of TRPA1 (Figure 3). Calcium flux in the TNF-α treated cells was significantly higher than in IL-1β-treated cells, likely due to the overall higher induction of TRPA1 expression by TNF-α. Therefore, it is possible that TRPA1 (not TRPV1) can be functionally involved in the calcium-induced responses of inflamed IVD cells. To verify the function of TRPA1/TRPV1, AITC was used to study gene and protein expression of inflammation and catabolic mediators in untreated as well as cytokine-treated IVD cells.

Cytokine untreated IVD cells (expressing TRPV1) were used to test the downstream effects of TRPV1 activation (*n* = 3–4). In unstimulated IVD cells, the gene expression of inflammation/pain mediators (IL-6, IL-8, NGF, COX-2) (Figure 4A–D) and most of the ECM remodeling enzymes (ADAMTS4, ADAMTS5, MMP3, TIMP1, TIMP2) was unchanged by AITC (Figure 4E–J). IL-6 and IL-8 concentration in culture media was close to the lower detection limit (~zero) (not shown). MMP1 was significantly induced by 10µM AITC (Figure 4H). The gene expression of COL1A1, COL2A1, and aggrecan in AITC-treated cells was not significantly different from the control (Figure 4K–M). Due to the fact that cytokine-untreated cells did not express TRPA1, the observed changes in MMP1 expression were not TRPA1-dependent. MMP1 upregulation by AITC may be an unrelated non-specific effect (as AITC did not induce calcium flux in untreated IVD cells).

IVD cells treated with pro-inflammatory cytokines (expressing TRPA1 and TRPV1) were used to test the downstream effects of TRPA1 activation (*n* = 3–4). In IL-1β-treated cells, AITC did not influence gene expression of inflammation mediators IL-6 and IL-8 (Figure 5A,D) and pain mediators NGF and COX-2 (Figure 5C,F). Interestingly, the protein release of IL-8 was significantly reduced by 3 μM AITC (Figure 5E). 10 μM AITC significantly reduced the gene expression of ADAMTS5 (Figure 5H), while MMP1 was induced (Figure 5J). Gene expression of other ECM remodeling enzymes (ADAMTS4, MMP3, TIMP1, TIMP2) and ECM genes (Figure 5M–O) in AITC + IL-1β-treated cells was not different from the IL-1β-only controls.

10 μM AITC significantly induced the gene and protein expression of IL-8 (Figure 6D,E) in TNF-α-treated cells (expressing TRPA1, reduced TRPV1). AITC did not influence the expression of IL-6 (Figure 6A,B), NGF, and COX-2 (Figure 6C,F). The gene expression of ADAMTS5 was significantly reduced (Figure 6H), while MMP1 was induced by 10 μM AITC in TNF-α-treated cells (Figure 6J). Gene expression of other ECM remodeling enzymes (MMP3, ADAMTS4, TIMP1, and TIMP2) was not different from TNF-α-treated control. Gene expression of COL1A1 was significantly reduced (Figure 6M), while the other tested ECM proteins (COL2A1 and aggrecan) were not influenced by AITC in TNF-α-treated cells. AITC did not affect the gene expression of TRPA1 itself (Appendix A).

### 2.5. Motion Segments of TRPA1 and TRPV1-Deficient Mice

As our in vitro study showed a possible involvement of TRPA1/TRPV1 in IVD metabolism, we also focused at their effects in vivo. The possible involvement of TRPA1 and TRPV1 in ECM homeostasis was studied by comparing the tail motion segments of young (two months old) and mature (seven months old) TRPA1 wild type (WT) and knock-out (KO) mice (*n* = 5 in each group) as well as young (four months old) and mature (seven months old) TRPV1 KO mice (*n* = 5 in each group). Anatomically, IVD structures from TRPA1 KO and TRPV1 KO mouse were intact with a distinctive central NP tissue, surrounded by lamella fibers of annulus fibrosus (AF) and sandwiched with cartilaginous endplates. However, FAST staining revealed a depletion of sulfated glycoproteins (Alcian blue) in the NP and a reduction of glycosaminoglycan (GAGs) (Safranin O) in the outer AF and vertebral growth plates of matured TRPA1 KO mice, when compared with TRPA1 WT matured controls. No discernible changes of GAG contents were detected in the young TRPA1 KO mouse IVD (Figure 7). On the contrary, no significant changes in the GAG contents were evidenced in the NP and vertebral growth plates of TRPV1 KO mice (Figure 8). The data suggested a functional importance of TRPA1 in GAG production during IVD maturation.

## 3. Discussion

Several TRP channels are expressed in joints and IVDs, but their potential biological function and therapeutic relevance are not fully understood. The first aim of this study was to investigate the expression of TRP channels in IVD cells in normal and inflammatory conditions, as inflammation is one of the major hallmarks of DDD. We showed that IL-1β significantly induced gene expression of TRPA1 and TRPV4 and reduced TRPC6. TNF-α significantly increased the gene expression of TRPA1 and TRPV2, while reducing TRPV1. It was previously reported that TRPA1 and TRPV1 are commonly expressed in sensory neurons that can innervate joints and IVDs as well as in chondrocytes, where they are associated with degenerative changes [24,27,28].

We found that gene expression of TRPA1 is undetectable in mature healthy human IVDs and untreated cultured IVD cells. Interestingly, TRPA1 was expressed in 20% of degenerated IVDs, possibly due to the presence of pro-inflammatory cytokines. TRPA1 was also expressed in cells isolated from fetal disc tissue and in healthy juvenile samples, which pointed towards its involvement in disc development and/or maturation. FAST staining of tail motion segments of TRPA1 KO and TRPV1 KO mice suggested that TRPA1 might be involved in the homeostasis of GAG maintenance during the development of the IVD. Although these results corresponded to our findings in human IVD tissues/cells, they should be interpreted with caution, as differences between mice and human IVDs exist (e.g., the presence of notochordal cells in mice, degenerative status of mature mouse disc vs. human disc).

To evaluate the possible effects of TRPA1/TRPV1 activation in IVD cells, we used the TRPA1/TRPV1 agonist allyl isothiocyanate (AITC) [29]. AITC (or mustard oil) is commonly regarded as pro-inflammatory and nociceptive [30]. For example, TRPA1-deficient mice do not display acute pain-related behavior after the application of AITC to paws [31]. Our data indicated that the activation of TRPA1 may be the main mechanism for AITC-evoked increase in [Ca^2+^]_i_ in IVD cells. In the non-inflamed IVD cells (expressing TRPV1), 10 µM AITC stimulation was not associated with significant pro-inflammatory/catabolic effects, except for an increase in MMP1. In contrast, AITC-mediated regulation of gene expression of ADAMTS5, IL-8, and COL1A1 in cytokine-stimulated cells was likely to be TRPA1 dependent. The TRPA1 downstream effects may depend on agonist concentration, as previously shown by others [32,33] and in our study (reduced IL-8 in cells treated with 3 μM AITC vs. upregulated IL-8 in in cells treated with 10 μM AITC). The expression level of TRPA1 itself can be another reason for the observed differences in IL-8 release between IL-1β and TNF-α treated cells (lower relative TRPA1 expression in IL-1β-stimulated cells vs. TNF-α-stimulated cells). The downregulation of COL1A1 and upregulation of IL-8 in TNF-α, but not IL-1β-treated cells, may be related to lower expression of TRPV1. Altogether, our data suggested that TRPA1 might be involved in the regulation of ECM homeostasis. However, major limitation of this study is low sample number, which prevents definite conclusions.

DDD is considered to be similar to chronic arthritis, due to the fact that common mechanisms are involved in the progression of both diseases [12]. Similar to our findings, the expression of TRPA1 in primary human osteoarthritic (OA) chondrocytes increased upon stimulation with IL-1β, IL-17, LPS, and resistin [24]. Horvath et al. (2016) showed that the markers of chronic arthritis (chronic mechanical hypersensitivity, joint swelling, histopathological alterations, vascular leakage) were significantly reduced in TRPA1 KO mice (vs. wt), which indicated the involvement of TRPA1 in this disease [34]. A similar association of TRPV1 with chronic arthritis was previously demonstrated [35,36]. Interestingly, acute joint pain behaviors were not modified in TRPA1 KO mice [34]. The distinct roles of TRPA1 in chronic vs. acute arthritis could be attributed to a different distribution of TRPA1 (and possibly TRPV1) on sensory nerves and non-neuronal cells in these pathological conditions [34], e.g., due to the presence of pro-inflammatory cytokines. In this context, the modulation of inflammation itself can possibly regulate TRP channel activities (e.g., TRPV1 can be sensitized/desensitized by endogenous products of inflammation [37]). Importantly, pro-inflammatory cytokines (IL-1β and TNF-α) can cause [Ca^2+^]_i_ increase in OA chondrocytes [34], but likely not in IVD cells [34], which might be related to differences in the TRP channel expression/activation/function in OA and DDD.

Chronic inflammation in both OA and DDD is associated with neuronal plasticity, which is an important mechanism in the development and maintenance of chronic pain [38]. Our current study did not employ an IVD degeneration/pain model, and thus it did not test the involvement of sensory neurons. Future studies will focus on the interplay between TRPA1/TRPV1 in inflamed IVD cells and DRG neurons, as well as on more specific activation/inhibition experiments by gene editing, both in vitro and in experimental animals.

Although TRPA1/TRPV1 antagonists/agonists have reached clinical trials for the treatment of inflammatory and neuropathic pain [39,40,41], discrepancies as to whether and how these channels contribute to the underlying mechanisms of inflammatory and neuropathic hypersensitivity can still be found in the literature [42]. Some endogenous ligands of TRPA1 might not yet be discovered and it is still unclear how physiological loading, which is an important parameter in IVD health, regulates the activity of TRPA1. It is likely that TRPA1 activation may be protective under certain circumstances and/or in particular cell types, possibly including the IVD. The protective anti-inflammatory effects of TRPA1 were recently demonstrated in mouse model of colitis [43], with TRPA1 KO mice having a significantly higher ‘Disease Activity Index’ and levels of pro-inflammatory neuropeptides and cytokines in the distal colon [43]. Another study showed that both the colonic and systemic administration of AITC and capsazepine (another TRPA1 agonist) induced a profound, body-wide TRPA1-mediated desensitization of nociception in mice [44]. The authors suggested that systemic desensitization through TRPA1 might provide a novel strategy for the medicinal treatment of various chronic inflammatory and pain states [44], which possibly included DDD.

Concerning other TRP channels that were regulated by an inflammatory environment in this study, increased TRPV4 expression/signaling in the IVD has been associated with decreased tissue osmolarity and the production of pro-inflammatory cytokines [17]. Our study provided evidence that IL-1β itself can regulate gene expression of TRPV4 in IVD cells., the Gene expression of TRPC6 was shown to be reduced in IVD cells under microgravity [18], but elevated in IVDs with increasing degeneration grade [19]. In our study, TRPC6 was downregulated by IL-1β treatment. To explain these inconsistencies and their pathophysiological relevance, the activity, stability, and subcellular localization of TRPC6 will be investigated in the future. Possibly, the activity of TRPC6 may be regulated by exocytosis [45], while cytoplasmic calcium may influence its expression and degradation [46,47], the levels of which are dysregulated in degenerated IVD cells [48]. To our knowledge, this is the first study that reported the downregulation of TRPV1 and upregulation of TRPV2 by TNF-α in IVD cells. The expression of TRPV2 was shown upregulated in inflamed DRGs [49], where it possibly participated in calcitonin gene-related peptide (CGRP) release [50].

## 4. Materials and Methods

### 4.1. Subjects

#### 4.1.1. Non-Degenerated Human IVD Tissue

cDNA was synthesized from the non-degenerated human IVD cells, (gift provided by Prof. Lisbet Haglund from the Department of Surgery at McGill University, Canada), and prepared as previously described [51]. Informed consent for tissue collection was obtained from family members and the study was approved through the local ethics committee (A04-M53-08B).

#### 4.1.2. Human IVD Tissue

25 human degenerated lumbar IVD samples were used for direct tissue analysis. These biopsies were obtained from 20 donors [mean age = 46.2 years (16–76 years); nine male and eleven female] undergoing elective spinal surgery. IVD samples were intraoperatively separated into annulus fibrosus (AF, *n* = 11) and nucleus pulposus (NP, *n* = 14), followed by macroscopic tissue evaluation. The assessment of the disease state was performed using Pfirrmann and Modic grading. Demographic details are shown in Table 3—Tissue and [19]. An additional 30 lumbar degenerated IVDs, removed during surgeries for disc herniation (DH) or degenerative disc disease (DDD), were used for preparation of primary cell cultures. All of the biopsies were obtained with patient’s informed consents. The Ethics Committee of Cantons Zurich and Lucerne approved the study (#1007). Demographic details are shown in Table 3—Cells.

#### 4.1.3. Human Fetal IVD Cells

The cell cultures were derived from biopsies that were obtained in accordance with the Ethics Committee of the University Hospital of Lausanne (Ethics Protocol 51/01) and following the Federal Transplantation Program guidelines. The cell banks are managed in the Department Biobank following the regulations of the Biobank for clinical research for both the fetal and adult tissue. Specific biopsies consisting of a spinal unit of three vertebra and two discs representing tissue of 6.5 mm × 5 mm in size were obtained from fetal tissue following the voluntary interruption of pregnancy at 14–16 weeks of gestation. Biopsies were first rinsed in 1% penicillin/strepotomycin in phosphate buffer saline (PBS) and adjacent soft tissue that was delicately dissected from the main disc tissue. We thereafter prepared one IVD and two adjacent vertebrae from a fetal spine unit; primary cultures used Dulbecco’s Modified Eagle’s medium (DMEM) (41966-029, Gibco, Waltham, MA, USA) that was supplemented with 10% fetal bovine serum and 100 mM l-Glutamine (25030-024, Gibco). Culture conditions were at 37 °C under 5% CO_2_. One juvenile patient was also used. Cell cultures from the juvenile patient following discectomy were established as above, except type II collagenase digestion, was implemented. Cells were expanded and stored frozen in liquid nitrogen at passage 1 or 2 and the cells were thawed, expanded in monolayer at passage 3, and used for analysis following passage 4. Table 3 shows donor details.

#### 4.1.4. Knock-Out Mice

Tails from C57BL/6 TRPA1 wild-type (WT) and knock out (KO) mice and 15 TRPV1 WT and KO mice were used (Table 4). The mice were divided into two groups: young (two, four months old) and mature (seven months old) and euthanized in the context of other research activities. C57BL/6 mice were use as control for TRPV1 KO as this strain was backcrossed 10× to the C57BL/6 background. Immediately after euthanasia with pentobarbital (100 mg/kg i.p.), the tails were dissected, skinned, rinsed in PBS, and fixed in 4% paraformaldehyde solution (Szkarabeusz Kft., Pecs, Hungary) in 0.1 M phosphate buffer. After two days, the fixed tails were washed with PBS and then placed into 10% EDTA (E6758, Sigma, St. Louis, MO, USA) exchanged every two days. After 10 days, the decalcified samples were washed in PBS and placed into 70% ethanol (51976, Sigma) at 4 °C until paraffin embedding. Paraffin blocks were then used to prepare 5 μm sections.

### 4.2. Cell Culture

#### 4.2.1. 2D Cell Culture

IVD tissue was diced to around 1 mm^3^ pieces and treated with a mixture of 0.3% dispase (04942078001, Roche Diagnostics, Mannheim, Germany) and 0.2% collagenase (17454, SERVA Electrophoresis, Heidelberg, Germany) in PBS at 37 °C for 4–8 h to isolate the cells. After the incubation, the suspension was filtered through a 70 µm cell strainer (542070, Greiner Bio-One, Kremsmünster, Austria), centrifuged at 700× *g* for 5 min, and resuspended in DMEM/F12 (31330-038, Thermo Fisher Scientific, Waltham, MA, USA) supplemented with 10% fetal calf serum (FCS, F7524, Merck, Darmstadt, Germany), 100 units/mL penicillin, 100 µg/mL streptomycin, and 250 ng/mL amphotericin B (15240-062, Gibco, Carlsbad, CA, USA). The primary IVD cells were expanded in monolayer in a standard cell culture incubator (5% CO_2_, 37 °C) up to passage 3 before being used in the experiments.

#### 4.2.2. 3D Cell Culture

The IVD cells were detached using 1.5% trypsin (15090-046, Thermo Fisher Scientific) and seeded in 1.2% alginate (71238-50G, Sigma, St. Louis, MO, USA) at a density of 4 × 10^6^ IVD cells per 1 mL alginate, as described previously [52]. Briefly, the cells-alginate mixture was dropped into 102 mM calcium chloride solution (1.02382, Merck) while using a sterile syringe and needle and left for 8 min to polymerize under gentle stirring until beads were formed. After washing with 0.9% NaCl (1.06404, Merck) and PBS (3 × 1 min), the beads were transferred into six well plates and pre-cultured for seven days.

#### 4.2.3. Cell Viability of 3D Cell Culture

Cell viability in the alginate beads was tested using calcein/ethidium homodimer staining. Staining solution was prepared by mixing culture media with 2 µM ethidium homodimer (46043, Sigma) and 2 µM calcein-AM (17783, Sigma). 200 µL/well of the staining solution was added into a 96-well plate containing beads (one bead per culture condition per well) and then incubated for 30 min. Subsequently, the beads were gently squeezed between cover slips and three photos were randomly captured with a fluorescence microscope (Olympus IX51, Tokyo, Japan) at the wavelength of 515 nm (calcein: living cells) and 620 nm (ethidium: dead cells). The cell numbers in each image were counted by ImageJ ver.1.51j8 and averaged. Cell viability is shown as the number of living cells per total cells.

### 4.3. Treatments

#### 4.3.1. 2D Cell Cultures

Table 5 shows all treatments. Experiments were conducted in culture media without antibiotics and FCS (= experimental media). For gene expression analysis and ELISA, the IVD cells were seeded into T25 flasks (3.5 × 10^5^ cells/flask) or six-well plates (3 × 10^5^ cells/well). For immunofluorescence, 1 × 10^5^ cells were seeded into the wells of chambered slides (155380, Thermo Fisher Scientific). For Calcium imaging, the 4 × 10^4^ cells were seeded into 96-well plates in triplicates and incubated for 18 h. The next day, complete media was changed to the experimental media. After 2 h in experimental media, the cells were exposed to 5 or 10 ng/mL recombinant TNF-α (315-01A, PeproTech, Umkirch, Germany) or IL-1β (200-01B, PeproTech) for 18 h. Non-stimulated cells were used as the controls. To investigate the effects of TRPA1 and TRPV1 activation, 3 and 10 µM allyl isothiocyanate (AITC, TRPA1 agonist, 377430, Sigma) was used either in untreated cells or in IL1-β and TNF-α treated cells (focus on TRPA1, whose expression is increased in an inflammatory environment). The EC50 value for AITC that is reported in the literature is approximately 3 μM, while AITC concentrations higher than 10 μM may cause channel desensitization [32,33].

#### 4.3.2. 3D Cell Cultures

The experiments were conducted in culture media without antibiotics and with FCS. On day 7, cells in alginate beads were stimulated with 5 ng/mL IL-1β and collected after 24 h (day 1), eight days (day 8), and 15 days (day 15) (Table 5). Culture media for the latter group was exchanged on day 8, with new 5 ng/mL IL-1β. At the end of the experiment, the cells were liberated from the beads in 1.9 mL of 55 mM sodium citrate solution (71406, Sigma) and centrifuged at 700× *g* for 5 min. Cell pellets were used for subsequent analyses.

### 4.4. Analyses

#### 4.4.1. Gene Expression Analysis of IVD Tissue

RNA extraction from IVD tissue and the following steps were performed according to [19]. For cDNA synthesis, two micrograms of RNA were used in a total volume of 60 µL, while using the reverse transcription kit (4374966, Applied Biosystems, Foster City, CA, USA). For samples with lower yields, the reverse transcription was conducted at reduced concentrations. cDNA (10 ng/well) was mixed with TaqMan Fast Universal PCR Master Mix and TaqMan primers (Table 6) to quantify gene expression. The obtained *C*_t_ values were analyzed by a comparative method and displayed as 2^−d*C*t^ values, with GAPDH as housekeeping gene.

#### 4.4.2. FAST Staining

To study the glycosaminoglycan (GAG) contents in IVD, a multi-dye histological staining using Fast green, Alcian blue, Saffranin-O, and tartrazine was performed on IVD tissue sections accordingly [53]. In brief, parafilm embedded tissue sections were first dewaxed in xylene and then rehydrated in a stepwise gradient of ethanol. The IVD sections were first stained with 1% Alcian blue 8GX (A3157, Sigma) pH 1.0 for 1min, followed by 0.1% Saffranin-O (S8884, Sigma) for 3 min. Saffranin-O reddish colour differentiation was performed in 25% ethanol for 15 s. The tissue sections were then stained in 0.08% Tartrazine (T0388, Sigma) with 0.25% acetic acid for 45 s and finally counterstained by 0.01% Fast green (F7258, Sigma) solution for 5 min. Sections were finally air-dried, mounted in DePeX (BDH Laboratory; Poole, UK), and examined under a Nikon Eclipse 80i microscope (Tokyo, Japan).

#### 4.4.3. Gene Expression Analysis of IVD Cells

RT-qPCR was performed to analyze the expression of target genes (Table 6). The cells were lysed with 1 mL Trizol (Genezol, GZR200, Geneaid biotech, New Taipei City, Taiwan) and RNA was isolated according to the manufacturer’s recommendations. Briefly, after adding chloroform (132950, Sigma), the samples were mixed well for 15 s, left for 5 min at RT, and centrifuged (12,000× *g* for 15 min). Supernatants were carefully transferred to new RNase free tubes, 500 µL of isopropanol (20842, VWR chemicals, Fontenay-sous-Bois, France) was added and mixed well. After 5 min, the samples were centrifuged again (12,000× *g* for 15 min). The pellets were washed with 75% ethanol (1.00983, Merck) at 7500× *g* for 5 min and then dissolved in RNase free water (10977, Invitrogen, Carlsbad, CA, USA). The purity and concentration of the resulting RNA were measured using the NanoDrop (ND-1000, Thermo Fisher Scientific). 1 µg of total RNA was reverse-transcribed to cDNA using a reverse transcription kit (4374966, Applied Biosystems, Foster City, CA, USA). qPCR of the mixture of primers/probes (Table 6) and master mix (4367846, Applied Biosystems) was performed on the CFX96 Real-Time System (Bio-Rad Laboratories, Hercules, CA, USA). The amplification program was as follows: heating at 95 °C for 10 min; 40 cycles of 95 °C for 1 s and 60 °C for 20 s. The relative expression level was calculated by the dd*C*_t_ method. For normalization purposes, the samples with undetectable expression were assigned *C*_t_ value 40. The results are shown as fold change, relative to control or relative to cytokine treatment.

#### 4.4.4. Immunofluorescence

The cells were seeded into the wells of chambered slides, washed with PBS, fixed with ice cold methanol (10 min), and blocked with 5% normal goat serum in PBS (1 h at RT). A primary antibody recognizing the N-terminus of the human TRPA1 protein (NB110-40763, Novus Biologicals) was diluted in 1% normal goat serum in PBS (1:200) (G9023, Sigma) and applied 1h under agitation at RT. Cells without primary antibody were used as immunospecificity control. After washing with PBS (3 × 5 min), a secondary antibody that was diluted in 1% normal goat serum (1:200) (Cy2 anti-rabbit IgG, 111-225-144, Jackson Immuno Research) was applied for 1 h at RT under agitation. Next, cells were washed with PBS (3 × 5 min), coverslipped with 1–2 drops of antifade mounting medium with DAPI (VECTASHIELD; H-1200), and imaged with a fluorescence microscope (Olympus IX51).

#### 4.4.5. Enzyme-Linked Immunosorbent Assay (ELISA)

To quantify the release of inflammatory markers from IVD cells, the cell culture medium was collected and analyzed with IL-6 and IL-8 ELISA kit, according to the manufacturer’s protocol (IL-6 555220, IL-8 555244, BD Biosciences, San Jose, CA, USA). 96-well plates were coated with capture antibody overnight. After washing, the wells were blocked in assay diluent, washed, loaded with samples or human recombinant IL-6 or IL-8 protein standards, and incubated for 2 h at RT. After washing, detection antibody and streptavidin-horseradish peroxidase (HRP) were applied for 1 h. Next, the plates were washed and substrate solution was added. After 30 min, the reaction was stopped by kit stop solution and the absorbance was measured at 450 nm, with 570 nm correction. The IL-6 and IL-8 concentrations were calculated based on the standard curve. IL-6 and IL-8 in culture media are shown relative to the cytokine treatment.

#### 4.4.6. [Ca^2+^]_i_ Imaging

Fura-2 QBT™ Calcium Kit was used to measure the increase in intracellular calcium (R8197, Molecular Devises, San Jose, CA, USA). Briefly, culture media was replaced with Fura-2-AM kit solution and the cells were incubated for 1 h. Basal Fura-2 fluorescence was recorded for 5 min using a plate reader (infinite M200 pro, Tecan, Männedorf, Switzerland) at an excitation wavelength of 340 and 380 nm and an emission wavelength of 510 nm. After five cycles, the cells were exposed in 100 µM of AITC (to activate disc cells within a short timeline) and the measurement was continued. Ionomycin (13909, Sigma) was used as a positive control for channel stimulation. Data is shown as the ratio of 340/380 wavelengths.

#### 4.4.7. Statistical Analysis

Statistical analysis was performed in GraphPad Prism 8.0.0. Groups were compared using the Kruskal–Wallis nonparametric test followed by Dunn’s multiple comparison test. Data is shown as mean ± SD. *p* < 0.05 was considered to be statistically significant (* *p* < 0.05, ** *p* < 0.01, *** *p* < 0.001).

## 5. Conclusions

To our knowledge, this is the first study that demonstrated a cytokine-dependent increase in the gene expression of TRPA1, TRPV2, and TRPV4 and a decrease in the gene expression of TRPC6 and TRPV1 in human IVD cells. Although TRPA1 and TRPV1 are commonly associated with inflammatory pain, their activation in inflamed IVD cells did not have profound pro-inflammatory and catabolic effects. Instead, TRPA1 expression and activation was associated with ECM metabolism. Future studies will use targeted gene editing techniques to elucidate the exact role of TRPA1/TRPV1 in DDD.

## Figures and Tables

**Figure 1 ijms-20-01767-f001:**
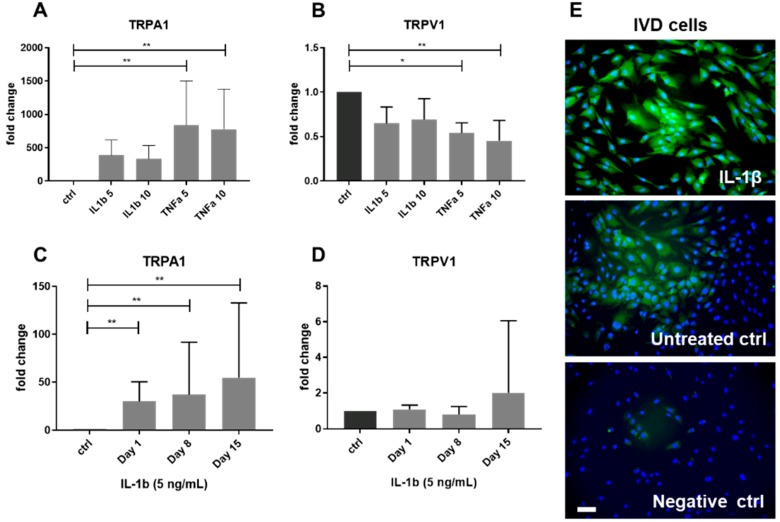
Gene expression of TRPA1 and TRPV1 in IVD cells treated with 5 and 10 ng/mL interleukin-1 beta (IL-1β) or tumor necrosis factor alpha (TNF-α). Gene expression of (**A**) TRPA1 and (**B**) TRPV1 in two-dimensional (2D) culture (Graph shows 2^−dd*C*t^ (mean ± SD, *n* = 5). Gene expression of (**C**) TRPA1 and (**D**) TRPV1 in IVD cells cultured in three-dimensional (3D) alginate beads and treated with IL-1β for 15 days. Graph shows 2^−dd*C*t^ (mean ± SD, *n* = 4–10). Asterisks indicate statistical significance (* *p* < 0.05, ** *p* < 0.01, Kruskal–Wallis test and Dunn’s multiple comparison test). (**E**) Protein expression of TRPA1 in IVD cells treated with IL-1β and untreated (DAPI = blue, TRPA1 = green). Negative control images show cells without secondary antibody. Scale bar is 50 μm.

**Figure 2 ijms-20-01767-f002:**
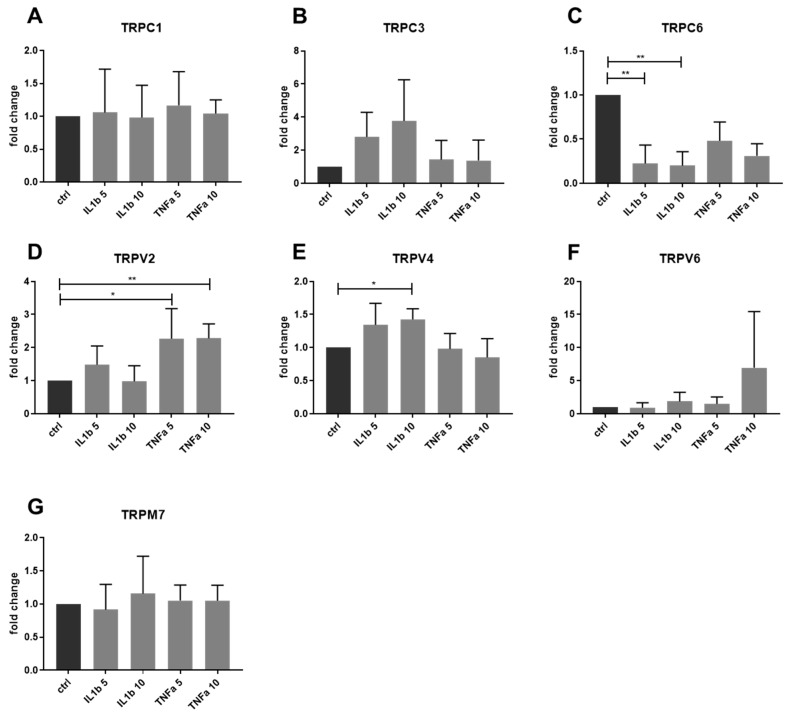
Gene expression of TRP channels in IVD cells treated with 5 and 10 ng/mL IL-1β or TNF-α. Gene expression of (**A**) TRPC1, (**B**) TRPC3, (**C**) TRPC6, (**D**) TRPV2, (**E**) TRPV4, (**F**) TRPV6, and (**G**) TRPM7. Graphs show 2^−dd*C*t^ (mean ± SD, *n* = 5). Asterisks indicate statistical significance (* *p* < 0.05, ** *p* < 0.01, Kruskal–Wallis test and Dunn’s multiple comparison test).

**Figure 3 ijms-20-01767-f003:**
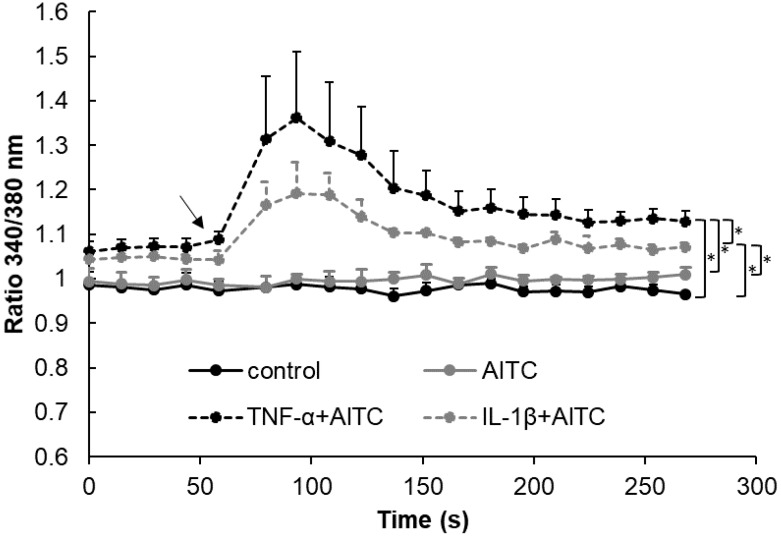
Calcium flux in IVD cells untreated (control) and treated with 100 μM Allyl isothiocyanate (AITC) with and without 10 ng/mL IL-1β or 10 ng/mL TNF-α. Graph shows calcium flux as 340/380 signal ratio (mean ± SD, *n* = 3). Asterisks indicate statistical significance (* *p* < 0.05, Kruskal–Wallis test and Dunn’s multiple comparison test). The arrow indicates AITC application.

**Figure 4 ijms-20-01767-f004:**
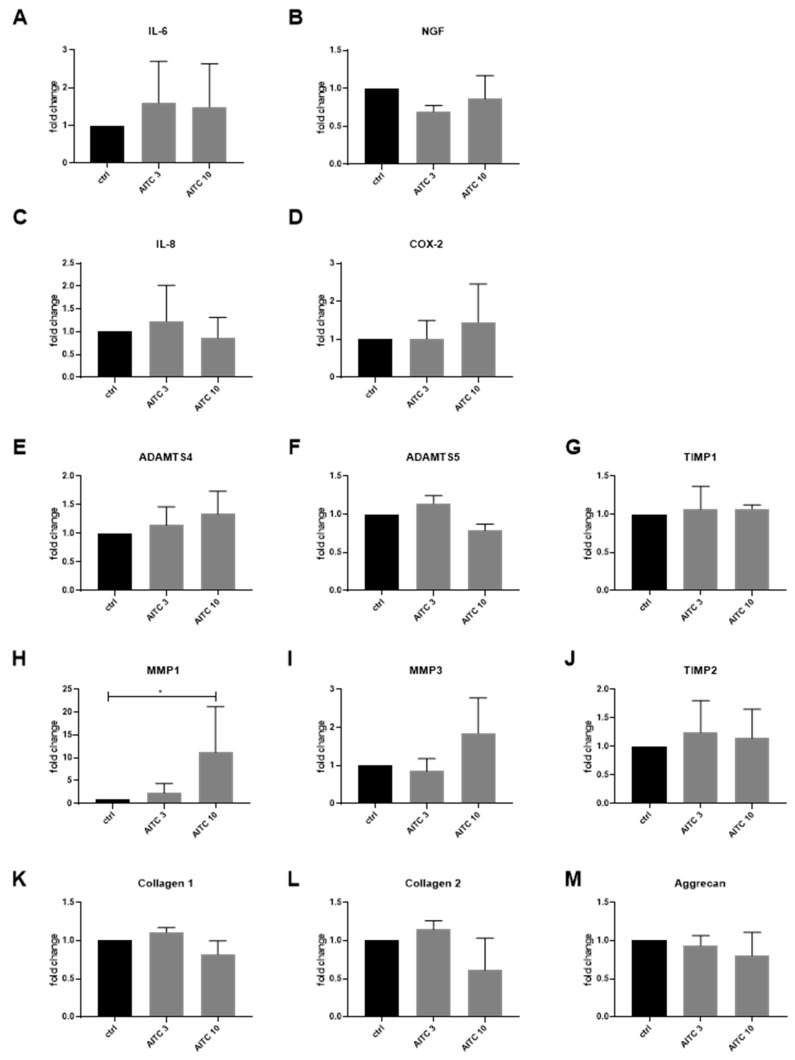
The effects of TRPA1 agonist allyl isothiocyanate (AITC) on the gene expression of inflammation markers and extracellular matrix (ECM) molecules in IVD cells without cytokine pre-treatment. Gene expression of (**A**) interleukin-6 (IL-6), (**B**) nerve growth factor (NGF), (**C**) interleukin 8 (IL-8), (**D**) cyclooxygenase-2 (COX-2), (**E**) ADAMTS4, (**F**) ADAMTS5, (**G**) tissue inhibitor of matrix metalloproteinase 1 (TIMP1), (**H**) matrix metalloproteinase 1 (MMP1), (**I**) matrix metalloproteinase 3 (MMP3), (**J**) tissue inhibitor of matrix metalloproteinase 2 (TIMP2), (**K**) COL1A1, (**L**) COL2A1, and (**M**) Aggrecan in IVD cells treated with 3 and 10 μM AITC. Graphs show gene expression and protein release calculated relative control (2^−dd*C*t^, mean ± SD, *n* = 3). Asterisks indicate statistical significance (* *p* < 0.05, Kruskal–Wallis test and Dunn’s multiple comparison test).

**Figure 5 ijms-20-01767-f005:**
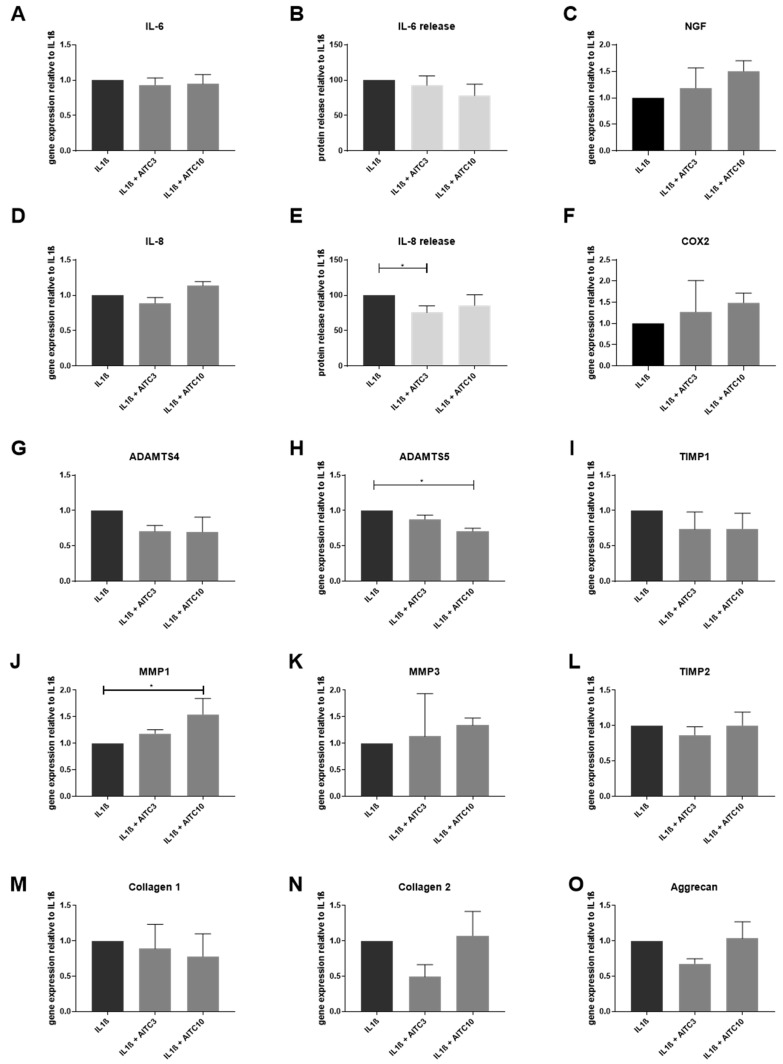
The effects of TRPA1 agonist allyl isothiocyanate (AITC) on the expression of inflammation markers and ECM molecules in IL-1β-treated cells. Gene expression of (**A**) IL-6 and (**D**) IL-8 in IVD cells treated with 10 ng/mL IL-1β ± 3 and 10 μM AITC. Protein release of (**B**) IL-6 and (**E**) IL-8 in IVD cells that were treated with 10 ng/mL IL-1β ± 3 and 10 μM AITC. Gene expression of (**C**) NGF, (**F**) COX-2, (**G**) ADAMTS4, (**H**) ADAMTS5, (**J**) MMP1, (**K**) MMP3, (**I**) TIMP1 and (**L**) TIMP2, (**M**) COL1A1, (**N**) COL2A1, and (**O**) Aggrecan in IVD cells that were treated with 10 ng/mL IL-1β ± 3 or 10 μM AITC. Graphs show gene expression and protein release calculated relative to IL-1β treatment (mean ± SD, *n* = 3–4). Asterisks indicate statistical significance (* *p* < 0.05, Kruskal–Wallis test and Dunn’s multiple comparison test).

**Figure 6 ijms-20-01767-f006:**
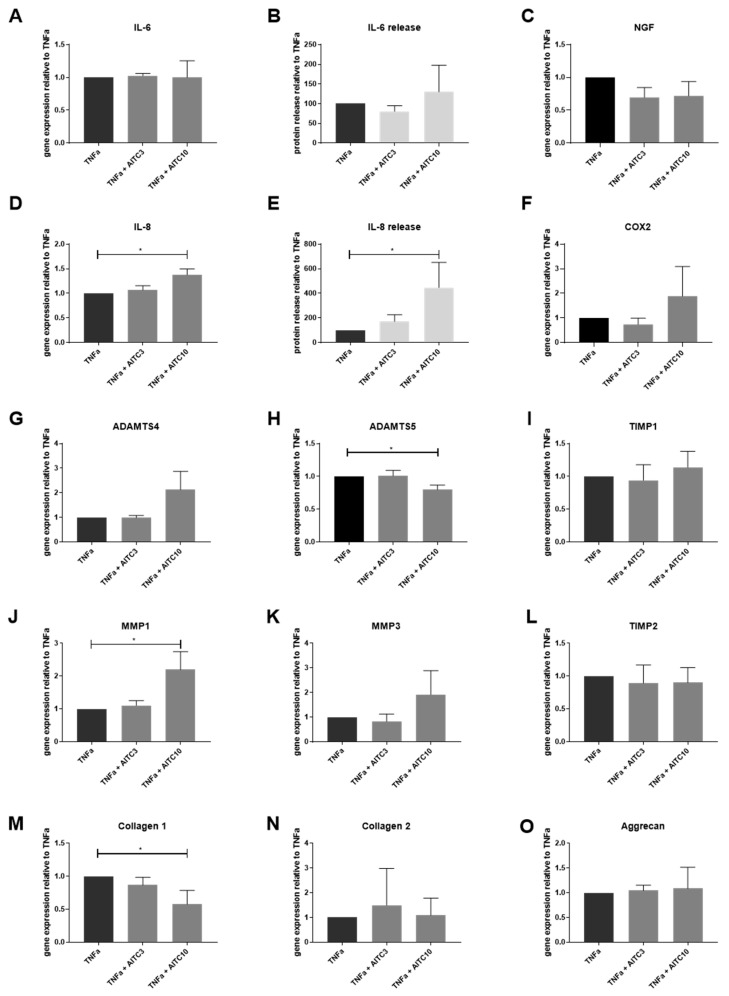
The effects of TRPA1 agonist allyl isothiocyanate (AITC) on gene expression of inflammation markers and ECM molecules in TNF-α-treated cells. Gene expression of (**A**) IL-6 and (**D**) IL-8 in IVD cells treated with 10 ng/mL TNF-α ± 3 or 10 μM AITC. Protein release of (**B**) IL-6 and (**E**) IL-8 in IVD cells treated with 10 ng/mL TNF-α ± 3 or 10 μM AITC. Gene expression of (**C**) NGF, (**F**) COX-2, (**G**) ADAMTS4, (**H**) ADAMTS5, (**I**) TIMP1, (**J**) MMP1, (**K**) MMP3, and (**L**) TIMP2, (**M**) COL1A1, (**N**) COL2A1, and (**O**) Aggrecan in IVD cells treated with 10 ng/mL TNF-α ± 3 and 10 μM AITC. Graphs show gene expression and protein release calculated relative to TNF-α treatment (mean ± SD, *n* = 3-4). Asterisks indicate statistical significance (* *p* < 0.05, Kruskal–Wallis test and Dunn’s multiple comparison test).

**Figure 7 ijms-20-01767-f007:**
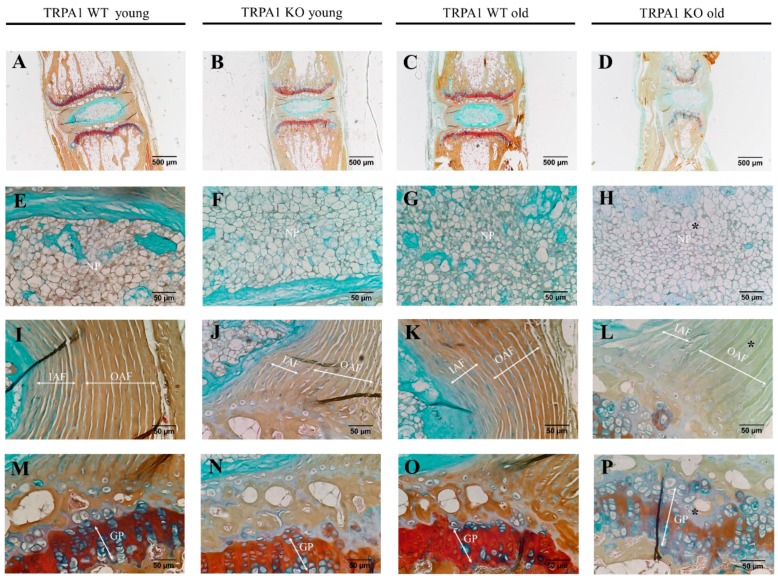
FAST staining of IVDs of TRPA1 wild-type (WT) and knock-out (KO) mice. The tail motion segments of TRPA1 young WT (**A**,**E**,**I**,**M**), TRPA1 young KO (**B**,**F**,**J**,**N**), TRPA1 old WT (**C**,**G**,**K**,**O**), and TRPA1 old KO (**D**,**H**,**L**,**P**) mice. The nucleus pulposus: NP (**E**–**H**); inner annulus fibrosus: IAF and outer annulus fibrosus: OAF (**I**–**L**); vertebral growth plate: GP (**M**–**P**) are also shown in higher magnification. Asterisks (*) indicate depletion of glycosaminoglycan deposition in IVD. Scale bars indicate 500 µm in upper panel (**A**–**D**), but 50 µm in lower panels (**E**–**P**).

**Figure 8 ijms-20-01767-f008:**
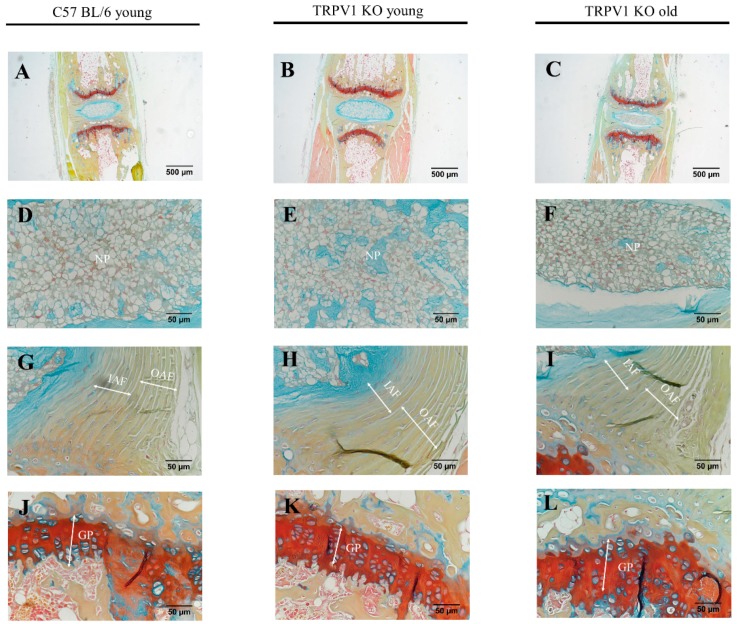
FAST staining of IVDs of TRPA1 WT and TRPV1 KO mice. The tail motion segments of C57 BL/6 young WT (**A**,**D**,**G**,**J**), TRPV1 young KO (**B**,**E**,**H**,**K**), and TRPV1 old KO (**C**,**F**,**I**,**L**) mice. The nucleus pulposus: NP (**D**–**F**); inner annulus fibrosus: IAF and outer annulus fibrosus: OAF (**G**–**I**); and, vertebral growth plate: GP (**J**–**L**) are also shown in higher magnification. Scale bars indicate 500 µm in upper panel (**A**–**C**), but 50 µm in lower panels (**D**–**L**).

**Table 1 ijms-20-01767-t001:** Gene expression of TRPA1 and TRPV1 in human intervertebral disc (IVD) tissue. Some donor tissues were divided into nucleus pulposus (NP) and annulus fibrosus (AF), resulting in more AF and NP samples (*n* = 21 in region, pain score, grade) than a total number of donors (*n* = 20).

Degenerated Lumbar IVDs	*n*	2^−^^d*C*t^ (Mean ± SD)	Region	Pain score	Grade	Age (Mean ± SD)
**TRPA1**	in 4 out of 20	0.0006 ± 0.001	2 in AF 2 in NP	2 pain score 2; 2 pain score 3	2 grade 2; 2 grade 3	60 ± 15.6 max: 76; min: 39
**TRPV1** [26]	19 out of 20	0.0047 ± 0.0024	10 in NP 9 in AF 2 in mix	6 in pain score 1; 11 in pain score 2; 4 in pain score 3	4 grade 2; 8 grade 3; 6 grade 4; 3 grade 5	54 ± 15 max: 80; min: 31

**Table 2 ijms-20-01767-t002:** Gene expression of TRPA1 and TRPV1 in untreated and treated human IVD cells (2D). *n* = 5 donors. Values show fold change relative to the untreated cells. In case of expression under the detection limit, d*C*_t_ was set at 40 cycles.

Treatment	Untreated	IL-1β 5 ng/mL	IL-1β 10 ng/mL	TNF-α 5 ng/mL	TNF-α 10 ng/mL
**TRPA1**	No expression	385.18 ± 233.06	333.42 ± 199.80	842.06 ± 659.98	780.23 ± 600.17
**TRPV1**	Expression	0.64 ± 0.18	0.69 ± 0.23	0.53 ± 0.11	0.45 ± 0.22

**Table 3 ijms-20-01767-t003:** Donors used for tissue and cell culture analyses. DH = herniation, DDD = degenerative disc disease, AF = annulus fibrosus, NP = nucleus pulposus, uk = unknown.

**Tissue**							
**Donor**	**Age**	**Gender**	**Pathology**	**Tissue**	**Level**	**Grade**	**Experiments**
T1	30	m	DDD	AF, NP	L4/5	II	qPCR
T2	46	f	DH	AF, NP	L5/S1	III	qPCR
T3	34	m	DH	AF, NP	L5/S1	III	qPCR
T4	46	f	DH	AF, NP	L4-S1	V	qPCR
T5	59	f	DDD	AF, NP	L5/S1	V	qPCR
T6	62	f	DDD	AF	L5/S1	V	qPCR
T7	66	f	DH	AF	L4/5	II	qPCR
T8	53	m	DH	NP	L5/S1	II	qPCR
T9	59	m	DH	NP	L4/L5	II	qPCR
T10	52	m	DDD	NP	L4/L5	III	qPCR
T11	64	f	DDD	NP	L4/L5	IV	qPCR
T12	76	f	DH	NP	L4/L5	III	qPCR
T13	16	f	DH	NP	L4/L5	III	qPCR
T14	31	m	DDD	AF	L4/5 L5/S1	IV	qPCR
T15	54	f	DH	NP	L5/S1	II	qPCR
T16	33	m	DH	AF	L5/S1	II	qPCR
T17	70	f	DDD	NP	L4/5	IV	qPCR
T18	39	m	DH	AF	L5/S1	III	qPCR
T19	28	m	DH	NP	L5/S1	II	qPCR
T20	21	f	DDD	AF	L4/5	III	qPCR
**Cells**							
**Donor**	**Age**	**Gender**	**Pathology**	**Tissue**	**Level**	**Grade**	**Experiments**
C1	44	F	uk	uk	L3/L4	uk	qPCR
C2	82	M	uk	uk	L5/S1	uk	qPCR
C3	28	M	uk	uk	L5/S1	uk	qPCR
C4	uk	uk	uk	uk	uk	uk	qPCR, ELISA
C5	uk	uk	uk	uk	uk	uk	qPCR, ELISA
C6	39	M	DDD, DH	Mix	L4/L5	IV	ELISA
C7	58	M	DDD, DH	Mix	L5/S1	IV	qPCR, ELISA
C8	46	F	DH	-	L5/S1	IV	qPCR, ELISA
C9	52	M	DDD, DH	Mix	L5/S1	V	qPCR, ELISA
C10	46	F	DDD, DH	Mix	L4/L5	IV	qPCR, Ca imaging
C11	58	M	uk	Mix	L4/L5	IV	Ca imaging
C12	31	M	DDD, DH	Mix	L5/S1	IV	qPCR
C13	46	M	DH	Mix	C5/C6	III	qPCR
C14	40	F	DH	Mix	L4/L5	III	qPCR
C15	40	M	DH	NP	L4/L5	III	qPCR
C16	66	F	DH	NP	L4/L5	III	Ca imaging
C17	uk	uk	uk	uk	uk	uk	qPCR, ELISA
C18	50	F	DH	Mix	L5/S1	IV	qPCR
C19	42	M	DH	Mix	L5/S1	V	qPCR
C20	68	F	listhesis	Mix	L4/L5	III	qPCR
C21	38	M	DH	Mix	L5/S1	III	qPCR, ELISA
C22	41	F	DH	Mix	L4/5	III	qPCR, ELISA
C23	42	M	DH	NP	L5/S1	IV	qPCR, ELISA
C24	41	F	DH	Mix	L5/S1	III	qPCR, ELISA
C25	45	M	DH	NP	L4/L5	IV	qPCR, ELISA
C26	71	uk	DDD	Mix	L4/5	III	qPCR, ELISA
C27	55	F	DH	NP	L5/6	I	qPCR, ELISA
C28	55	F	DH	NP	L5/6	II	Immuno
C29	55	M	DH	Mix	L5/S1	II	Immuno
C30	58	M	DH	Mix	L4/5	IV	Immuno
C31	34	M	-	NP, AF	L1/2-L2/3-L3/4	I	qPCR
C32	55	F	-	NP, AF	L1/2	III	qPCR
C33	52	M	-	NP	L1-L5	I	qPCR
C34	17	M	-	NP	T12-S1	I	qPCR
LC1	16	M	-	mix	uk	II	qPCR
LFC2	Fetal IVD cells, Male, 16 weeks: p5						qPCR
LFC3	Fetal IVD cells, Male, 14 weeks: p4						qPCR
LFC4	Fetal IVD cells, Male, 14 weeks: p4						qPCR

**Table 4 ijms-20-01767-t004:** Mouse spines used for FAST staining.

Mice	2 Months Old	4 Months Old	7 Months Old
TRPA1 WT	*n* = 5	-	*n* = 5
TRPA1 KO	*n* = 5	-	*n* = 5
TRPV1 KO	-	*n* = 5	*n* = 5
C57BL/6	-	*n* = 5	-

**Table 5 ijms-20-01767-t005:** Details of cell culture treatments.

Compound	Catalog Number	Function	Concentration	Exposure Time	Experiment
TNF-α	315-01A PeproTech	Inflammatory cytokine	5, 10 ng/mL	18 h	qPCR Ca imaging
IL-1β	200-01B PeproTech	Inflammatory cytokine	5, 10 ng/mL	18 h	qPCR Ca imaging
			5 ng/mL	1, 8, 15 days	qPCR (from 3D)
			5 ng/mL	18 h	Immuno
AITC	377430 Sigma	TRPA1 agonist	3 μM, 10 μM	18 h	qPCR, ELISA
			100 μM	during the measurement	Ca imaging

**Table 6 ijms-20-01767-t006:** Target genes and assay identification (ID) numbers of corresponding TaqMan primers (TaqMan Gene Expression Assays; Thermo Fisher Scientific). TRP = transient receptor potential; MMP = matrix metalloproteinase; TIMP = tissue inhibitor of matrix metalloproteinase; ADAMTS = a disintegrin and metalloproteinase with thrombospondin motifs; COX-2 = cyclooxygenase-2; NGF = nerve growth factor. IL-6 = interleukin 6; IL-8 = interleukin 8; HKG = housekeeping gene.

Target Gene	Assay ID	Putative Association with Inflammation
TBP	Hs00427620_m1	HKG in the cell culture study and fetal cells
GAPDH	Hs02758991_g1	HKG in the tissue study
TRPA1	Hs00175798_m1	inflammatory pain [12]
TRPC1	Hs00608195_m1	bladder inflammation (neuronal) [54]
TRPC3	Hs00162985_m1	inflammatory pain [55]
TRPC6	Hs00988479_m1	IVD inflammation (putative) [18,19]
TRPV1	Hs00218912_m1	neuroinflammation [20]
TRPV2	Hs00901648_m1	inflammation in DRG [49]
TRPV4	Hs01099348_m1	lung inflammation [56]
TRPV6	Hs00367960_m1	association with TNF-α [57]
TRPM2	Hs01066091_m1	inflammatory and neuropathic pain [58]
TRPM7	Hs00559080_m1	inflammation in colitis [59]
IL-6	Hs00174131_m1	inflammation mediator [60]
IL-8	Hs00174103_m1	inflammation mediator [60]
MMP1	Hs00233958_m1	cleaves mainly collagens (I, II, III) [61]
MMP3	Hs00968305_m1	cleaves proteoglycans and collagens (II, III) [61]
ADAMTS4	Hs00192708_m1	cleaves mainly aggrecan [61]
ADAMTS5	Hs01095518_m1	cleaves mainly aggrecan [61]
TIMP1	Hs00234278_m1	inhibits MMPs (1, 3) and ADAMTS (4) [61]
TIMP2	Hs01092512_g1	inhibits all MMPs [61]
COX-2	Hs00153133_m1	pain mediator [62]
NGF	Hs00171458_m1	nerve ingrowth [38]
COL2A1	Hs00264051_m1	ECM constituent
COL1A1	Hs00164004_m1	ECM constituent
Aggrecan	Hs00153936_m1	ECM constituent

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
