# Peer review of "Expression and Activity of TRPA1 and TRPV1 in the Intervertebral Disc: Association with Inflammation and Matrix Remodeling"

_ijms, 2019, doi:10.3390/ijms20071767_

Reviewer 1 Report

In this paper, the authors demonstrated a cytokine-dependent increase in TRPA1 gene expression and its association with matrix homeostasis. The results are clearly presented and the conclusions are objectively drawn based on their experimental results.

I have few comments about their manuscript:

Comments:

1. The authors referred to results on TRPV1 expression in IVD cells as previously demonstrated  (reference 25). However, reference 25 is a submitted, yet-to-be peer-reviewed manuscript that should not be referenced here as established and/or proven evidence.

2. Table 4 (line 98), that should be identified as Table 1 instead (please see minor points below) describes 4 patients as TRPA1 positive, which corresponds to the description in the text (lines 89-90). However, pain score and grade include 5 patients (e.g.: 2 pain score 2; 3 pain score 3). The same happens for TRPV1, where pain score and grade are described for 21 patients (pain score: 6+11+4 = 21; grade: 4+8+6+3 = 21). Should this be 4 instead? 

3. In the abstract (Line 37), the authors described that inflamed IVD cells are activated by 10 microM AITC. However, AITC was only used at 100 microM for intracellular calcium imaging experiments. Since the involvement of TRPA1 and TRPV1 was tested using concentrations of 3 and 10 microM, it would be preferable and relevant for this study if the authors show that these lower concentrations (3 and 10 microM AITC) activate IVD cells.

Minor comments:

1. Figure legends and section 4.4.7: Indicate the post-test used after the Kruskal-Wallis.

2. Line 90-91: Define AF and NP after first used.

3. Table numbering should be corrected. There are two Table 4 in the text (page 3 and page 16).

4. Table 5, line 124: Indicate the units of the numbers in the table. Or describe these in the Table legend.

5. Figure 3: Indicate the time frame of AITC application.

6. Figures 7 and 8: Include scale bar description. The scale bar embedded in the images are illegible.

7. Line 259: The original paper (PMID: 21315593) should be cited, instead of reference [51]

8. Line 295: There are certainly endogenous agonists for TRPA1 (PMID: 24756722). This sentence should be edited accordingly.

9. Correct subscripts and superscripts throughout the manuscripts. (e.g.: lines 351, 370, 381, 399, etc).

10. Line 487: micro symbol is missing

11. Table 1 (page 12): Grade scores for tissues and cells are defined with different notations. Unknown is defined as 'u' in the table heading but used as 'uk', 'Uk' or '-' in the table. 

Reviewer 2 Report

.    Q1:  Line 53-54: “Disc injury or degenerative disc disease (DDD), a progressive multifactorial disorder of the intervertebral disc (IVD),” This section sounds to me that Disc injury or DDD and IVD are the same. If so, line 55-58 you are trying to explaining the mechanism of the proinfammatory substances including interleukin-1 beta (IL-1β), tumor necrosis factor alpha (TNF-α), prostaglandins and proteases that further promote degradation of extracellular matrix (ECM) and the release of neuropeptides . However, line 58-61: In the IVD, the pro-inflammatory cytokines IL-1β and TNF-α can act directly as nociceptive triggers, but also induce expression of other potentially nociceptive molecules, such as interleukin-6 (IL-6) or  interleukin-8 (IL-8) [6]. 
These sentences sound to me that Disc or DDD and IVD are different. If these diseases are the same, why you used two long sentences to describe the same? If these are different, why in the line 53-53 your implication that these are the same?  

.    
Q2: From the context, you still do not provide enough evidence to support why you chose TRPA1 and TRPV1 as targets for LBP. Would it be possible you can show us that from disc you have done the RT-PCR and the results show that the expression levels of mRNA of TRPA1 and TRPV1 are the highest among them, and therefore you go straight toward to them? 

.    Q3: line 105 you need to explain the predilections of TNF-alpha, instead of IL-1beta to TRPV1  

.    Q4: line 109: the reason you chose for 15 days? 

.    Q5: line 254:  why TRPA1 isolated from fetal disc tissue 
and only 20% TRPA1 could be detected in adults? It is not reasonable that you contribute to these changes to the development. If so, TRPA1 should be detected in all maturation disc, shouldn’t it?

And you mentioned that The “abundance” of TRPA1 itself can be 
another reason for observed differences in IL-8 release 
in line 267? This inspection is far away from the previous sentences. Would it be possible that the genetic differences in different subjects lead to different expression? 

Q6: I totally agree with your line 273-285, because I also noted that the different DRG from different mouse expressed different TRP channels, and even the DRGS from the same mouse showed different TRP expressions, and therefore I can understand your pain. However, this is not the point. Could your problem may be due to the proinflammatory substances enhanced TRP channels expression levels and functions? You should add a sentence or a functional study to show that anti-proinflammatoion may neutralize the effects of TRPs. 

Author Response

Round  2

Reviewer 1 Report

The authors have adequately addressed the comments.

Please, provide the correct reference in line 268. Reference 51 is a review/book chapter and not the original paper showing that TRPA1-deficient mice do not display acute pain-related behaviour after application of AITC to paws [51]. As said before, this was shown in Everaerts et al. 2011 (PMID: 21315593).

The manuscript is acceptable for publication.

Author Response

Thank you for your comment, we added the reference:

To evaluate the possible effects of TRPA1/TRPV1 activation in IVD cells, we used the TRPA1/TRPV1 agonist allyl isothiocyanate (AITC) [50]. AITC (or mustard oil) is commonly regarded as pro-inflammatory and nociceptive [3]. For example, TRPA1-deficient mice do not display acute pain-related behavior after application of AITC to paws [3][51].

[3] W. Everaerts, M. Gees, Y.A. Alpizar, R. Farre, C. Leten, A. Apetrei, I. Dewachter, F. van Leuven, R. Vennekens, D. De Ridder, B. Nilius, T. Voets, K. Talavera, The Capsaicin Receptor TRPV1 Is a Crucial Mediator of the Noxious Effects of Mustard Oil, Current Biology, 21 (2011) 316-321.

The reference numbering is currently wrong, as the submission system removed our endnote links. We asked the editorial office to restore the endnote links in the next version. We will make sure during the proofreading that the reference is numbered correctly.
